# Adapting Reverse Mentoring Strategy to SMEs: A New Pilot Model Implemented in Brazil

Dafna Schwartz [1,*], Raphael Bar-El [2,3] and David J. Bentolila [4]

1   Adelson School of Entrepreneurship, Reichman University, P.O. Box 167, Herzliya 46150, Israel
2   Public Policy & Administration Department, Ben Gurion University of the Negev, Beer-Sheva 8410501, Israel; rbarel@bgu.ac.il
3   Department of Applied Economics and Administration, Sapir Academic College, Shaar Hanegev 7956000, Israel
4   Logistics and Global Supply Chain Program, Ruppin Academic Center, Emek Hefer 4025000, Israel; davidb@bento.co.il
*   Correspondence: dafna.schwartz@idc.ac.il; Tel.: +972-528844466

**Abstract:** In recent years, the Reverse Mentoring (RM) model has gained popularity in large companies. Although the prevailing RM model—junior employees mentoring senior employees—benefits both groups and promotes innovation, small and medium-sized enterprises (SMEs) do not implement it due to lack of economies of scale, organizational capacities, and skilled junior employees. We devise a new RM model for SMEs that overcomes these disadvantages. First, the intervention of an intermediate, trusted professional entity initiates and supports the program for several companies; second, the mentors are not junior employees, but external graduate students with education in innovation. A pilot experiment was tested in the state of Ceara, Brazil. The preliminary findings support the new model's feasibility and efficacy for SMEs. The intervention stimulated significant innovative ideas and resulted in out-of-the-box thinking, identification of potentials for innovation opportunities, and adaptation of an open innovation approach, which is important for SMEs with limited financial and non-financial resources. This study contributes to the literature on SMEs and RM by offering a new model that can overcome existing market failures experienced by SMEs. Empirical testing demonstrates its feasibility.

**Keywords:** SME; reverse mentoring; innovation; open innovation; knowledge management; technological change; Brazil

## 1. Introduction

Companies are trying to respond quickly to innovative technology and trends by supporting innovation and adapting their business strategy and work plan to the rapidly changing business and technology environment. To achieve this, it is crucial for companies to ensure their competence with innovative technologies and trends, and have skilled employees who are able to adapt quickly and align with the company's strategy and plans [1,2].

Many companies implement mentoring programs in the workplace, where senior employees mentor and train less experienced employees. One mentoring program model that has gained popularity in recent years, primarily among large companies, is the Reverse Mentoring (RM) model. In the traditional mentoring approach [3–9], older, more experienced employee mentors, act as role models, and demonstrate commitment and high standards to junior employees who are considered the "novice" learners [3–5,10,11]. However, in the RM model, the traditional mentoring relationship is flipped, and the juniors play an active role in providing valuable knowledge to their senior counterparts. The juniors share updated technology, perspectives, and forthcoming trends with senior employees, who, in turn, share their experience with them. Although the motivational processes in

RM unfold differently for the two groups—junior and senior employees—involved in the exchange [11], both benefit from the knowledge transformation.

In RM, there is a two-way relationship between juniors and seniors who are jointly engaged towards a common goal of support and mutual learning for skill development [5]. This interaction infuses the organization with up-to-date perspectives, ideas, and technology to promote innovation. It also helps the organization adapt its strategy to the evolving business and technological environment with lower cost and effort [3–9]. However, it appears that the RM model is mostly implemented by large companies and not by small and medium enterprises (SMEs), even though the latter could benefit from it. This gap is the result of SMEs having weaker access to networks, knowledge sources, and relevant services, which is a reflection of a market failure. In many cases, the relative shortage of young, highly educated workers also contributes to the inability of SMEs to benefit from the advantages of new innovation trends, improving markets, and research advancements.

Apart from the gap in the empirical implementation of RM between large companies and SMEs, a gap also exists in the literature; most articles on RM that have been published in recent years are related to issues that are relevant to big companies, while those concerning SMEs have been relatively neglected. This research gap is especially concerning given that most of the companies in business are SMEs. The case of the state of Ceara in Brazil, which is considered in this article, is a good representation of this gap, as SMEs constitute a major element of Ceara's economy.

The intention of this study is to find a response to the market failure by elaborating on and conducting a pilot test for a new approach to RM. The current study contributes to the research on SMEs and RM in four ways:

(a)　Adds to the scant literature on the RM model;
(b)　Presents a new RM model for SMEs to overcome existing market failures;
(c)　Evaluates feasibility of RM for SMEs by presenting an experimental pilot study;
(d)　Demonstrates the contribution of RM to SMEs in terms of promoting innovation and enabling them to adapt their strategies to changes in the environment.

The remainder of this paper is structured as follows. Section 2 provides a short literature review of the general idea of RM. Section 3 presents the theoretical framework and proposes a new RM model and its relevance to SMEs. Section 4 describes the methodology and the characteristics of the firms that participated in the pilot project. Section 5 discusses and analyzes the findings. Section 6 presents the discussion and policy implications. Finally, Section 7 presents the study limitations and recommendations for further research.

## 2. Literature Review

A review of the RM literature shows that, in addition to sharing learning and updating senior employees, companies try to gain additional benefits through the program, such as increased diversity and the retention of talented employees. Although the RM model is implemented through various programs that vary in scale and focus (e.g., recruiting and retaining talented employees, technological updates, and identifying new markets), its ultimate goal is to coordinate shared learning between colleagues of diverse backgrounds to create symbiotic corporate learning. It aims to help the organizations in their efforts to promote innovation and gain competitive advantage [12] in the rapidly changing business and technological environment.

A study conducted in Poland [13] that interviewed participants of an RM program showed that the program is an efficient tool for multi-generational interaction and the sharing of knowledge, technology, and perspectives. The study demonstrated that the interaction between senior and junior employees aided in developing capabilities on both sides. On one hand, having been exposed to new ways of thinking, trends, and markets, the senior employees developed new skills by learning new technologies, being exposed to new ways of thinking, trends, and markets, and gain a feeling of greater relevance. On the other hand, the juniors benefit by learning from the experience of the seniors.

Comparable results were found in a study by Tomlinson [14]; based on semi-structured interviews with 16 pairs of participants in RM, the study reported that building trust, chemistry, and communication between junior employees and veterans improved inter-generational relationships. The program also assisted in developing the capabilities of both sets of employees. The seniors benefitted by learning from the mentor's up-to-date knowledge, technological expertise, and generational perspective. This improved their technological capabilities and ability to understand current and future trends, and fostered in them a feeling of greater relevance [7,9,10,13–15].

RM can also help decrease social isolation among older adults, by improving their digital competence and developing their intergenerational connections [15]. A study that analyzed a case study of an RM initiative for digital transformation in a large metal multi-national company based in India showed that it helped generate organizational cognitive change. The program encouraged building routines that enabled processes that facilitate learning and cognitive change [16].

Although the academic literature provides few details regarding the RM program's operational aspects, most of what has been written about it can be gleaned from its practi-tioners. Some examples of such documentation are discussed in the following sections.

## 2.1. Multi-Generational Interaction, Knowledge Sharing, and Developing Capabilities

Jack Welch, the former CEO of General Electric, was among the first to coin the term RM and to adopt the approach as early as 1999. The idea behind the original program was that younger employees would share information and knowledge with executives in the hope that the latter would teach the former about technological advances and tools [5,12].

Nestlé Germany also embraced the RM model in 2015 with its program called "Good Luck." The program paired young IT workers with company executives who had limited digital skills. According to the company's reports, this helped bridge the digital knowledge gap of veteran workers and enabled them to understand different and intergenerational perspectives [17]. The RM program also helped junior employees improve their non-technical (soft) skills because of their interaction with senior employees. This was also one of the benefits of the RM program that was implemented by Microsoft Norway [18].

At Aflac, an insurance company from the U.S., veteran IT professionals were paired with new employees who had just graduated to get them out of their comfort zones, and facilitate the transfer of knowledge of trends and best practices [12].

## 2.2. Retaining Millennial Employees and Fostering Diversity and Inclusion

Millennials seek attributes such as sense of value, visibility, and recognition; companies that fail to provide these often must contend with high turnover rates. RM programs can help companies in this context [10,19,20] as it empowers young employees, gives them a sense of self-realization, helps them build their network within the company, and exposes them to new career opportunities [20].

BNY Mellon's Pershing, a financial service company [10], implemented a successful RM program to achieve the goal of retaining millennial generation employees [12]. The program contributed to employee retention in the company by increasing the retention rate to 96% for the first cohort of millennial mentors. Nestlé Germany's RM program helped to increase the visibility of junior employees within the company and gave them a sense of being valued [14]. Estée Lauder launched an RM program to target millennials and Gen Z markets [21].

RM programs also assist in fostering diversity and inclusion [3]. In this respect, the mentors are not necessarily young employees, but employees from diverse backgrounds. The pairing could be between populations that are diverse in age, exceptionalities, ethnicity, and gender. The interaction between executives and senior employees with employees from different backgrounds can transform the company culture, reduce unconscious biases, and enhance the representation of this group at the C level, that is, executives at the highest management level [12].

RM programs were implemented at various universities in the UK with the aim of advancing women in general and certain ethnic groups in senior academic positions [22]. According to Wingard [12], companies that developed RM programs to create diversity and promote the inclusion of minorities, such as ethnic minorities, include Linklaters—a worldwide legal firm—and PricewaterhouseCoopers (PwC). Cisco adopted the RM model as early as 2011 to improve awareness of diversity in society [23]. Procter & Gamble Global launched an RM program aiming to increase diversity for women in senior positions [12]. They also launched another program in 2018 designed to increase the share of employees with special needs by connecting veteran workers with young workers with disabilities. Launched in 2014, PwC's Australian RM program was part of its diversity and inclusion agenda [24].

### 2.3. Driving Culture Change

The culture of a company is often shaped from the top down. RM offers the employees at the bottom of the hierarchy the opportunity to express their ideas, norms, and perspectives to senior employees, and thereby influence the company culture and strategy. This was the case with the RM program conducted at Estée Lauder. Millennial mentors not only taught senior executives the importance of social media for marketing, but also developed Dreamspace—a knowledge-sharing portal for exchanging ideas among company employees—which influenced the company culture [10].

As opposed to the one-way top–down approach within the hierarchy in traditional mentoring, this bottom–up and top–down interaction can contribute to organizations in the health sector, as pointed out by Raza [25]. It can facilitate change and raise the standard of healthcare leadership in the healthcare sector, where local healthcare leaders play a significant role in shaping organizational cultures.

### 2.4. Key Success Factors

Studies have attempted to identify the key factors that make RM programs successful. They show that it is important to have the "right match" between the mentors and mentees to establish trust, address mentees' fear and distrust, and ensure that they will not hesitate to reveal their lack of knowledge to junior employees [10]. It is also important to consult with mentees before finalizing the pairing [26] and to devote adequate time to achieving a healthy and fruitful mentoring relationship [13].

Other factors include ensuring strong commitment from the mentees [10], adequate level of engagement in the mentor/mentee relationship, support from the organization by means of engagement and support from officers and top management [5,13], and assistance from the human resources department to help alleviate difficulties in managing different generations within an organization [4]. Other factors related to contextual dimensions [27] are supportive of organizational culture [13], atmospheres conducive to cooperation [13], and a learning environment at all levels that breaks down barriers of status and position [5].

## 3. The State of Ceara

Ceara is located in the northeastern part of Brazil. It has a population of approximately nine million inhabitants. It is one of the poorer states in Brazil, recording the 5th lowest GDP per capita among the 26 Brazilian states. Despite its low GDP, the state has overcome adverse socioeconomic conditions and has experienced the largest increase in the national education quality index in both primary and lower secondary education since 2005 [28]. Manufacturing is the most significant area in the industrial sector, followed by civil construction and public utility industrial services. In 2010, the industrial sector of Ceara represented 23.7% of the state's economy, and agribusiness and services accounted for 4.2% and 72.1%, respectively [2].

Since 1950, the Industry Federation of the State of Ceara (FIEC), an institution linked to the National Confederation of Industry (CNI), has actively participated in the economic growth, expansion, and modernization processes of Ceara, taking on institutional

and political representation among diversified sectors. FIEC brings together 39 sectoral associations and representatives of several industrial production segments. It also stimulates the implementation of actions that support the industry by providing services in the fields of technology, entrepreneurial strategies, economic studies, and research, and aiding the qualification of entrepreneurs and their employees [29]. The current pilot study was part of the comprehensive measures undertaken by the FIEC from 2011 to mid-2014 to support the development of an innovation ecosystem that would foster innovation in industry and provide concrete measures to assist companies in developing their own innovation capabilities.

In 2011, the FIEC established a special unit, called UNIEMPRE, designed to function as a catalyst for the innovation ecosystem by stimulating links between the players in the ecosystem—industry, academia, and the government. It provides information and knowledge, assists firms in developing their innovation capabilities, develops the milieu's innovation capabilities, and influences the government's channels of intervention [1,29]. In this framework, one of the programs that was elaborated and implemented by the FIEC to assist companies, including SMEs, in developing their innovative capacity and skills was the new Innovation Agent (IA) program.

## 4. SMEs and the RM Model

### 4.1. Current Model

SMEs are extremely important for economic growth and social inclusion in Brazil, as shown by the Organisation for Economic and Co-operation Development (OECD) [30]. They account for 62% of the total employment and 50% of the national value added. However, there are wider productivity gaps in Brazil between SMEs and large companies than in the OECD area, which can be attributed to the low innovation level of Brazilian SMEs. SMEs are regarded as one of the main engines of economic growth and are the main source of new jobs [31,32]. However, despite their vital role in the economy, studies show that SMEs face a variety of barriers in promoting innovation [33–36], which is essential for their development and sustainability [37].

Despite difficulties in promoting innovation and searching for tools to overcome these difficulties, it appears that SMEs are not implementing the current RM program. Instead, the RM model is mostly being implemented by large firms, generally at medium or high technology levels. We could not find any relevant literature about RM in SMEs, most likely due to the scarcity of this phenomenon. We assume that the lack of RM programs in SMEs is not a result of rational market behavior (or the inefficiency of RM in the context of SMEs), but a distorted process linked to the specific characteristics of SMEs. Thus, exogenous support is expected to lead to more efficient business behavior, increasing the profitability of SMEs and their contribution to macroeconomic growth.

The smaller size of SMEs, mainly those in traditional industries, is a disadvantage when implementing RM programs. They lack the resources, both organizational and managerial [34,38], and the human capital of junior employees who can partake in the program [39,40]. Some of these disadvantages also exist in larger firms in traditional industries. These disadvantages create market failures that hinder the participation of SMEs and some larger firms in the current RM program.

This study presents a new RM model that can close this gap. Given the above constraints and potential contribution of RM to innovation and economic development, a new approach to RM was developed, proposing a model that overcomes the disadvantages that come with SMEs' size, scarcity of skilled employees, and limited organizational capabilities. The new RM model—the IA program—was tested as an experimental pilot project in Ceara, Brazil, and implemented by the FIEC.

### 4.2. New Model: Experimental Pilot Project

The new RM model aims to respond to market failures by adding components to the program that overcomes the deficiencies of SMEs in comparison to large, more advanced

companies. To overcome limits in organizational capabilities and size, it was suggested that the program not be executed by each individual company, but by an external intermediate professional organization.

The FIEC served as an intermediate professional trusted organization that executed the program for several companies simultaneously, creating economies of scale. An advisory committee was appointed by the FIEC, which included its representatives, external experts, and the authors of the current study. To address the shortage of skilled, up-to-date junior employees in SMEs outstanding graduates with a specialization in innovation studies from all over Brazil were recruited as mentors (Figure 1).

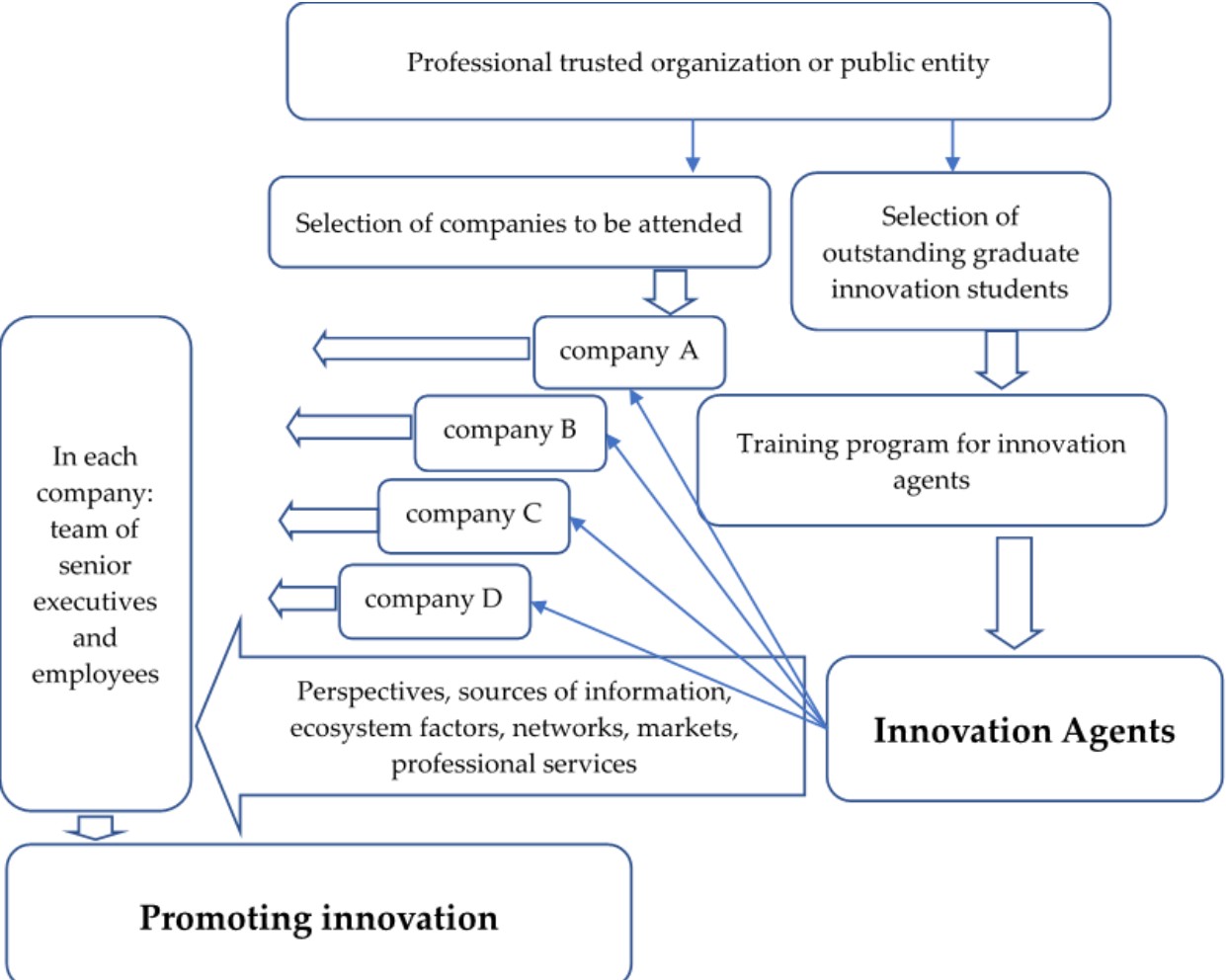

**Figure 1.** Schematic illustration of the new RM model.

It was emphasized to the graduates that their role as mentors was not to provide recommendations about specific professional activities, but rather to serve as an innovation agent by providing new ideas, technology, and perspectives. The graduates received three days of preliminary training by the FIEC in collaboration with a local university. The training covered relevant issues in the field of innovation, methodology to assess the potential for innovation in the company: information regarding public and private innovation ecosystem entities; access to regional, national, and international networks; data sources; financial options; research activities; and so on.

The graduates (the IAs) were expected to provide senior employees and executives with new knowledge on advanced tools and their implementation pertaining to their work. They were also expected to act as mediators for innovation ecosystem factors in Ceara by connecting the firms with universities, government innovation programs, and other firms facing similar problems. Their tasks included enhancing senior executives' awareness of

the importance of innovation; exposing industrialists to new ideas, perspectives, sources of information, ways of thinking, and modern technology and trends; instilling in senior employees a renewed sense of passion and energy; helping them identify potential innovation and assisting with its implementation; exploiting potential external entities (private and public) that could help the company in their innovation efforts, including support programs, and helping the company by approaching these entities.

The IA program was operated using the outreach approach, where the IAs approached the senior executives in their company. Two IAs were placed in each company where they were paired with senior executive(s). In some cases, the company appointed an innovation team for the mission. Meetings between the agents and the innovation team (one or more senior executives) took place on the company's premises and covered a wide range of topics. The agenda for each meeting was relatively flexible, but the IAs ensured that it aligned with the program guidelines and the company's expectations. The IAs had to submit a report to the FIEC, which included firm characteristics (e.g., size, location), the demand for innovation reflecting the direction that was defined for innovation, and the assistance needed from the IAs.

It should be noted that this new model applies to SMEs, not to micro-enterprises, which are too small to have the appropriate conditions to absorb such processes. These types of businesses are fully handled by the business owner, who makes all the decisions. As noted in previous studies [34,35], SMEs, especially those that have more than a few workers, have the capacity to absorb and promote innovation processes. Consequently, the pilot project did not include micro-enterprises.

### 4.3. Company Characteristics

For this pilot program, 22 enterprises were selected; we have detailed reports for 12 of them. Table 1 presents the characteristics of the 12 companies according to area, location, and size.

**Table 1.** The selected companies: size, sector, and location (Fortaleza/periphery).

| Number of Employees | Size | Sector | Location |
|---|---|---|---|
| 1074 | L | Dairy products | Periphery |
| 1400 | L | Shoes | Periphery |
| 250 | M | Mechanical metal | Periphery |
| 72 | M | Ceramics | Periphery |
| 70 | M | Confection | Periphery |
| 70 | M | Mechanical metal | Periphery |
| 50 | S | Mining | Periphery |
| 25 | S | Mechanical metal | Periphery |
| 30 | S | Ceramics | Fortaleza |
| 14 | S | Ceramics | Fortaleza |
| 10 | S | Dairy products | Periphery |
| 8 | S | Ceramics | Periphery |

Most of the companies (10) were located in a peripheral region (Periphery) (Baixo Jaguaribe); only two companies were in the Fortaleza metropolitan region (Fortaleza), the capital city of Ceara. All of them operated in traditional sectors. The companies were divided into three size categories: small enterprises employing up to 49 employees (S) (micro-enterprises generally employ up to five workers), medium-sized enterprises employing 50–249 employees (M), and large enterprises employing 250 or more people (L). Size was determined according to the number of employees based on the OECD's definitions [41]. As shown in Table 1, of the SMEs under the IA program, 50% are small companies (S) 33% are medium-sized companies (M), and only 17% are large companies (L).

## 5. Analysis and Results

### 5.1. Identifying Potential Areas for Innovation

Each company, together with the IAs, identified directions for promoting innovation and areas in management they need assistance in. Table 2 presents the new directions that were identified and the demand for support in management areas.

**Table 2.** Companies by size, directions for promoting innovation, and demand for support.

| Size | New Directions and Demand for Support |
|---|---|
| L | • Measurement system for the milk receiving tank volume;<br>• Production planning and control;<br>• Robotization for the AT3 Flex line—stacking and palletizing tetra pack boxes at the exit;<br>• Mass volume measurement system at the point of receipt of in-natura milk. |
| L | • Elaboration of studies to attract a supplier of technical and dyed fabrics;<br>• Waste management: patchwork processing;<br>• First Line Supervisor Course. HR training. |
| M | • Formation of scale in springs. The oven's atmospheric control system would solve the problem;<br>• Heat treatment oil filtration system (due to the large-scale formation);<br>• Robotic automation in the removal of springs after tempering. |
| M | • Automatic press feeding system;<br>• Grids for transport and roof drying;<br>• Use of heat from the tunnels and the generation of energy from the heat of the ovens;<br>• Preparation of a clean development mechanism project for obtaining carbon credits;<br>• Analysis laboratory with quality control of raw material clay;<br>• Implementation of production planning and control. |
| M | • Financial situational diagnosis;<br>• Franchise ability study;<br>• Consulting in intimate fashion design with pattern-maker training;<br>• Conducting a leadership development process and reducing absenteeism to act effectively in people management;<br>• Production planning and control; |
| M | • Expansion project review (new factory);<br>• Part injection with polypropylene. |
| S | • Acoustic insulation at the exhaust outlet. Possibility of generating energy with pressurized air;<br>• Productivity: increasing classifier efficiency;<br>• Marketing—New segments/new products: product for shrimp farm nurseries; introduce Okyta to the shrimp market. |
| S | • Planning to produce 24,000 kits/month; production capacity analysis;<br>• Marketing plan—how and where to sell;<br>• Personnel management plan for this purpose. This is an HR management plan; how to train the employee from the beginning;<br>• Use of crown production residue in the production of pine nuts;<br>• On some sprocket models, deflectors are welded to the sprockets to prevent accidents should the chain come loose. Deflectors are purchased from a company in São Paulo and represent approximately 7% of the company's revenue. |
| S | • Reformulation and launch of new products, preparing a strategic marketing plan;<br>• Definition of target market. Distribution channels; communication channels after sales. |
| S | • Expansion of sales of the current product portfolio and development of new products and markets. |
| S | • HR management: improving the recruitment process; developing tools and processes: the adoption of a policy of goals; responsibility matrix; conduct systematic team meetings;<br>• Marketing: new sales channels to avoid competition with informals. |
| S | • Identify and develop product for in-line production;<br>• Company strategic planning. |

We divided the newly identified directions into three categories of demand for promoting innovation: product, process, and marketing. In addition, we added two categories that refer to the demand for support in implementing those recommendations: in-the-field management or tools to adapt human resources (HR) to new directions (Table 3).

**Table 3.** Companies by size, areas of demand for innovation and for managerial and HR support.

| Company Size | Areas of Demand for Innovation | | | Areas of Demand for Support | |
|---|---|---|---|---|---|
| Size | Product | Process | Marketing | Managerial | HR |
| L |  | + |  | + |  |
| L | + | + |  | + | + |
| M |  | + |  |  |  |
| M |  | + |  | + |  |
| M | + | + |  | + | + |
| M |  | + |  | + |  |
| S | + | + | + |  |  |
| S | + | + | + | + | + |
| S | + | + |  | + |  |
| S | + |  | + | + | + |
| S |  |  | + | + | + |
| S | + |  |  | + |  |
|  | 7 | 9 | 4 | 10 | 5 |

The demand for support in management includes three sub-categories: planning, such as support in company strategic planning and "production planning and control"; demand for support in marketing and financing, including a variety of areas, such as financial diagnosis, identification of target markets, distribution channels, communication channels after sales, developing marketing plans.

The demand for support in HR includes: "supervisor courses, improving the recruitment process, and conducting systematic team meetings.

The findings show that for all the companies, the IAs assisted in identifying potential areas for innovation. For 50% of the companies, the IAs discovered at least two potential areas for promoting innovation. The dominant areas that were identified were related to processes (75%), such as recycling, logistics, increased productivity, and energy alternatives, and 60% were related to new markets or new products. These findings show that the recommendations were beyond incremental innovation, but are new options that can create a real change for companies. In some cases, it was to improve or change the production process and find new markets for expansion. In other cases, it was to develop a new product and find a market for it, and create a training program for employees to ensure that they are ready to implement such changes.

Grouping the companies by size showed that innovation in the production process was crucial for all medium-sized companies. The medium- and large-sized companies (except one) required managerial tools to implement the new strategies. It is important to note that most of the companies, including SMEs, wanted the IAs to help them with new recommendations, be it with regard to managerial tools, such as diagnostics, feasibility study strategic planning, and measures. Some companies, including SMEs (5 of 12 companies) also asked for measures in training employees to adapt to the changing environment.

*5.2. Adopting the Open Innovation Approach*

The open innovation concept was introduced by Chesbrough, who defined it in terms of purposive inflows and outflows of knowledge [36,37]. To be competitive and overcome limitations in size and resources, SMEs must interact with the external entities of the ecosystem by implementing an open innovation strategy [42–44], which can provide them with access to resources and assist them in promoting innovation [42,43,45,46]. The findings show that all companies asked the agents for assistance in approaching other entities of the

ecosystem to help them implement new directions that were identified for exploration and assist them in managerial areas. The number of external entities and their characteristics by company size are presented in Table 4.

**Table 4.** Identified external entities to approach for the implementation of new directions.

| Size | No. of New Directions | External Entities |
|------|:-----------------------:|-------------------|
| L | 4 | SENAI<br>IEL–RETEC<br>SENAI<br>SENAI |
| L | 3 | Industrial intelligence<br>External consultant<br>SENAI |
| M | 3 | SENAI<br>SENAI<br>SENAI |
| M | 6 | SENAI<br>SENAI<br>SENAI<br>NewLimp<br>SENAI<br>IEL–RETEC |
| M | 5 | External consultant<br>External consultant<br>SENAI<br>External consultant<br>IEL–RETEC |
| M | 2 | IEL–RETEC<br>SENAI |
| S | 3 | External consultant<br>External consultant<br>External consultant |
| S | 4 | IEL–RETEC<br>External consultant<br>SENAI MARACANAU<br>SENAI MARACANAU |
| S | 2 | External Consultant<br>External Consultant |
| S | 1 | IEL–RETEC |

The IAs assisted the companies in identifying and referring to two or more external entities, including professional consultants, public services, and public support programs. Stimulating the interaction of companies with other ecosystem entities is an important means of catching up to innovation processes [47] and leads to the adoption of open innovation strategies. Instead of only relying on their own capabilities, SMEs can take advantage of other ecosystem entities.

The IAs identified the best entity for interaction for each proposed area in the company. This could be an external private expert and/or a support organization. Within the support organization, they recommended a relevant support program. In many cases, the IAs also helped in implementing the programs and making initial contact. External entities include entities from the private and public sectors. From the public sector, there are two entities: SENAI and IEL-RETEC. The SENAI is the National Service for Industrial Training. It is a network of professional schools established and maintained by the Brazilian Confederation

of Industry. IEL–RETEC helps stimulate the technological advancement of companies and provides information necessary for the feasibility of projects.

Most of the SMEs requested assistance in approaching the public support entities SENAI or RETEC. This is crucial for SMEs, especially small enterprises, as they avoid approaching public support services because of the lack of familiarity with the programs or the lack of trust in their contribution.

## 6. Discussion: The Program in Retrospective

### 6.1. Contribution to SMEs

The results of the pilot program provide a clear indication of the existence of a viable response to the problem of SMEs' adoption of the RM model and can, therefore, help reduce the gap between them and bigger firms. Although adapting the model to the specific conditions of SMEs through the recruitment of graduate students in the process does not necessarily address all constraints in the pilot project, the study clearly shows that the RM model can make an important contribution in many areas. Moreover, even a partial success can lead to further efforts to improve the model and amplify the project.

In 2021, we conducted a focus group to gain an in-depth understanding of the program's contribution a few years after its implementation. The focus group consisted of representatives of entities that participated in the program. It included two IAs, one company CEO, FIEC representatives, and the advisory committee for the program. The overall feedback of all participants in the focus group was that the program made a significant positive contribution to the participating enterprises. It contributed to the companies by stimulating and encouraging them to think beyond their day-to-day activities. The program exposed them to contemporary trends, technologies, markets, and perspectives that they were not previously aware of and helped them promote innovation. The program was especially beneficial to SMEs that, as shown by previous studies, encounter difficulties in promoting innovation, especially regarding their production and organizational processes and expansion to new markets. An IA who took part in the discussion gave several examples of the program's contributions. One example was that, through the program, one company was able to identify a new potential market and gain entry into it, as well as expand its activities. Another example was the case of a factory that dealt with motorcycle equipment. As a result of the RM program, it was able to develop new products and increase its number of employees from 40 to 70. The CEO of said company (R.S.) participated in the focus group, and said, "The program was fundamental, both for me personally and for the company's growth." It also enhanced the company's processes by increasing the involvement of employees in the innovation process and rewarding them for their contributions.

Another contribution of the program was the applicability of the RM model to the academic setting, which is an untapped resource in the academy [48]. It is well known that one of the difficulties of graduates is the transition from university to the labor market without industry experience. Higher education institutions are facing increasing demands to develop employability skills as part of the learning process [49]. Hence, it would be a strategic directive to more effectively improve the transition of graduates from education to work [50,51]. Based on the focus group discussion, the graduates benefitted from the program by gaining industry experience, expanding their business networks, and being supported in their entry to the labor market. It also helped to strengthen the linkages between industry and academia.

### 6.2. Theoretical and Practical Implications

Theoretical research has put quite a considerable focus on the issue of the development of SMEs, trying to address their disadvantages in comparison with bigger enterprises. Many aspects were broadly covered, such as the question of the access to financial resources, scarcity of skills, and restricted networks. The issue of integration into the innovation process received much less attention. The results of this pilot project indicate the existence

of a few additional research fields that require further theoretical research: the adaptation of RM to the specific characteristics of SMEs as an important instrument for their integration into the global innovation process, and its relative contribution to firms with different sizes, types of activities, and ecosystem environments.

An important practical implication of the program is the need to adapt policy measures to the new needs of SMEs. One is the elaboration of support instruments for the adoption of an open innovation approach by SMEs. An example illustrated in this project is the support for the identification of other entities of the ecosystem that may contribute to the productivity of the SMEs. This is mostly important for SMEs with limited financial and non-financial resources. Another practical implication is the need for academic institutions to become more involved in the world of business. As found in this preliminary study, such involvement benefits both the graduate students in the process of integration into the business activity, and the SMEs, which were provided with more updated information and fresh ideas. The third and most important practical implication is the need for the involvement of public organizations as a response to the existence of a market failure. As explained above, the RM initiative cannot be implemented by every small or even medium enterprise (as it is taken by bigger enterprises). In any case of market failure, it should be performed by an external institution for the benefit of the enterprises and the national economy. In this case, the initiative was taken by an industrial association, but this may be performed by public government entities as well.

## 7. Conclusions

The RM model is more common among large companies for promoting innovation and adapting their strategy and workplan to the rapidly changing technology and business environment. In traditional mentoring programs, the more senior employees mentor and train the more junior employees. They help new employees to quickly absorb the organization's cultural and social norms and learn what they need to know to succeed in their job and role. Contrary to traditional mentoring, in the RM model, the juniors and less experienced employees function as mentors to more experienced employees.

This is a two-way interaction. While the juniors share updated knowledge, technology, and trends with the senior employees, the senior employees share their accumulated business experience. This interaction creates a mutual benefit for both the mentor and mentee. Senior employees are introduced to contemporary trends, technologies, and markets. They are also exposed to what the younger generation is interested in, and how the company can make itself attractive to them—as potential employees and as an important customer segment. The juniors benefit from the senior experience network and the opportunity to have an impact on company strategy and direction.

Although the RM model has increased in popularity in recent years, SMEs surprisingly do not implement the current model. Analyzing the literature and documents of practitioners, we identify that the root of the market failure is that the current RM model is not appropriate for SMEs as it does not respond to their specific characteristics. Size disadvantages, lack of highly qualified junior employees, and limited organizational capacity inhibit SMEs from implementing the RM model.

This study makes a modest contribution to the literature on SMEs and RM by presenting a new RM model called "The Innovation Agent (IA) program," adapted to SME characteristics and empirically evaluated in the state of Ceara, Brazil. This new RM model overcomes the market failures for SMEs by adding two components. The first is the intervention of an intermediate trusted professional entity (FIEC) that initiated and implemented the program for several companies. The second component is that the mentors are not junior employees, but external university graduate students, called "Innovation Agents," recruited by the FIEC especially for this mission.

Previous studies on SMEs show that the difficulties of SMEs in obtaining financial resources hinder their innovation potential. The current study shows that assistance through the proposed RM model can encourage and promote innovation among SMEs. The

program stimulated and encouraged companies to think beyond their day-to-day activities and decide on their strategy and new directions. All the companies identified new areas for promoting innovation, such as the development of new products and production processes, introduction to new markets, and the development of innovative marketing processes. In many cases, innovation was made in more than one area. The RM program also stimulated companies to consider ways to implement these new strategies by requesting support for managerial tools that can facilitate the implementation.

Although the new RM model responds to constraints encountered by SMEs, some of its features may be considered and analyzed by larger companies as well. Introducing graduate students with up-to-date knowledge of innovation to big companies may bring added value as they can offer new and fresh perspectives.

## 8. Limitations and Recommendations for Further Research

This pilot study was based on an experiment conducted on a few non-representative companies. The limited number of companies, as well as the limited representation of broader and more diverse business activities, imposed significant constraints on the use of appropriate quantitative statistical analyses. An additional important limitation is the time restriction imposed on the pilot project; an in-depth analysis of the new RM model requires a follow-up evaluation of the outcome of the experiment, which may be visible only after a few months or more in terms of increasing productivity, expanding production, introduction to new markets, and so on.

We believe that the model of an external intervention by a public institution recruiting young mentors with appropriate knowledge in innovation deserves more in-depth research. The case of SMEs in Brazil already provides some interesting insights and indications for further research. Specific industrial branches, the importance of local and regional innovation ecosystems, and the efficiency of alternative policy measures are critical issues that can be explored in more detail in future statistical research.

Another aspect that should be further investigated is the contribution of the new RM model to the employability of university graduate students. Employability is an important key factor in higher education, and in many cases the responsibility for employability has been transferred to higher education institutions [52].

**Author Contributions:** Conceptualization, methodology, analysis, writing: D.S., R.B.-E. and D.J.B. All authors have equal contributions. All authors have read and agreed to the published version of the manuscript.

**Funding:** This research received no external funding.

**Institutional Review Board Statement:** Not applicable.

**Informed Consent Statement:** Not applicable.

**Acknowledgments:** We are deeply thankful to the leaders of the Federation of Industries of Ceara (FIEC) (Federacao das Industrias do Estado do Ceará), especially to Roberto Proença de Macêdo and his team, to Carlos Matos Lima and to his team within the UNIEMPRE program. Special thanks to to Pedro Sisnando Leite and to Mônica Clark Cavalcante who provided an invaluable professional and personal support, and of course to all participants in this project.

**Conflicts of Interest:** The authors declare no conflict of interests.

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
