# Peer review of "Adapting Reverse Mentoring Strategy to SMEs: A New Pilot Model Implemented in Brazil"

_sustainability, doi:10.3390/su14159515_

Round 1
Reviewer 1 Report
The paper has been improved according to reviewer comments. It is rich, reflective and add new insights to research and practice. I recommend publication.
Author Response
Thank you very much.
Reviewer 2 Report
This version is improved compared to the previous version. The authors of the article properly handled all my comments.
Author Response
Thank you very much.
Reviewer 3 Report
1) Please add some lines showing the organization of the study at the end of the introduction section.
2) Please include the questions of the study in the introduction section.
3) Please format the lines 92-98 as per the style of MDPI.
4) Can the authors have a pictorial representation of the proposed model to make it easy for the reader to clearly understand the concept of the study and proposed model?
5) While I appreciate the improvement made to the article in the literature review I was still expecting a clear and concise answer to the question of how can the proposed model be applied to small or micro-enterprises? These are very small enterprises with a limited number of employees so how can we have senior people there.
6) I also expected to have a separate section for the implication of the study both theoretical and practical implications.
7) While I believe that the authors have a good background in the research I was shocked to see no limitations in the study.
8) The author should clearly distinguish between the implications and the discussion of the study as the discussion part seems to be including everything.
Kind regards
Ali

Author Response
Thank you for reviewing our manuscript. All the comments are valuable and very helpful for revising and improving our paper. A major revision of the paper has been carried out to take all of them into account. Therefore, we have sent the revised manuscript, titled “final revised version”, and a version containing all the changes to be visible, titled “revised version -track changes”.
At the following, the points mentioned by the reviewer will be discussed:
1) Please add some lines showing the organization of the study at the end of the introduction section.
Response:
Thank you for your comment. A paragraph has been added at the end of the introduction, as required.
2) Please include the questions of the study in the introduction section.
Response:
The questions of the study have now been included in the introduction.
3) Please format the lines 92-98 as per the style of MDPI.
Response:
Said lines (first paragraph of section 2.1) have been appropriately formatted as per the style of MDPI.
4) Can the authors have a pictorial representation of the proposed model to make it easy for the reader to clearly understand the concept of the study and proposed model?
Response:
We have included Figure 1 in the manuscript (section 4.2) to schematically present the proposed RM model.
5) While I appreciate the improvement made to the article in the literature review I was still expecting a clear and concise answer to the question of how can the proposed model be applied to small or micro-enterprises? These are very small enterprises with a limited number of employees so how can we have senior people there.
Response:
We fully agree with this comment and added a clarification in section 4.2 that the model may be applicable only to SMEs and not to micro-enterprises (the no. of section 4.2 was updated to reflect the revised revision in the manuscript- it was previously section 3). Furthermore, we show in Table 1 that the smallest enterprise included in the pilot project had eight workers, with a quite relevant distribution of occupations.
6) I also expected to have a separate section for the implication of the study both theoretical and practical implications.
Response:
We now present the theoretical and practical applications of the study in a separate section in section 6.2.
7) While I believe that the authors have a good background in the research, I was shocked to see no limitations in the study.
Response:
In Section 8, we have now introduced an additional paragraph describing the study’s main limitations, followed by suggested future research directions.
8) The author should clearly distinguish between the implications and the discussion of the study as the discussion part seems to be including everything.
Response:
Section 6.1 presents the discussion, while section 6.2 has been entirely dedicated to the implications of the study.

Reviewer 4 Report
There are several things that can be input in this article:
1. The author must be able to explain maximally, many paragraphs are written too short, so they need another explanation, or sentences can be put together in the previous or next paragraph.
2. Improvements in the making of tables, must be done according to the standards of the latest scientific publications, please correct them.
3. The use of style in the reference manager, both in the citation address and the bibliography must be appropriate. whether to use IEEE or Vancouver.
The full results of the review are written directly in the article script

Author Response
Thank you for reviewing our manuscript. All the comments are valuable and very helpful for revising and improving our paper. A major revision of the paper has been carried out to take all of them into account. Therefore, we have sent the revised manuscript, titles “final revised version”, and a version containing all the changes to be visible’ titles “revised version -track changes”.
At the following, the points mentioned by the reviewers will be discussed:
- The author must be able to explain maximally, many paragraphs are written too short, so they need another explanation, or sentences can be put together in the previous or next paragraph.
Response:
Thank you for your comment. We have revised the manuscript to reflect longer paragraphs that elaborate or provide more explanations on the topic being discussed. We have combined statements and paragraphs, especially in the introduction, literature review, and SMEs and the new model section to improve the overall structure of the text.
- Improvements in the making of tables, must be done according to the standards of the latest scientific publications, please correct them.
Response:
These changes have been incorporated as per your comment.
- The use of style in the reference manager, both in the citation address and the bibliography must be appropriate. whether to use IEEE or Vancouver.
Response:
The references have been edited following the journal's referencing style
Comments in the text:
The numbering in the introduction was corrected, references 1 and 2 were added at the beginning of the paragraph.
Quote corrected in section 2.2.
Authors name deleted after table 4.
The Conclusion part was shortened, implications included in another part.
Following the comments of reviewer 2 and reviewer 3, a last section is now “Limitations and Recommendations for Further Research”.
References corrected for the IEEE style.

Round 2
Reviewer 3 Report
Satisfied.
This manuscript is a resubmission of an earlier submission. The following is a list of the peer review reports and author responses from that submission.
Round 1
Reviewer 1 Report
The article makes an important contribution to the long-standing issue of business development of SMEs and of the constraints they face as a consequence of their relatively small size. Extensive research has already been done regarding the issues of access to finance, of participation in the process of innovation, of access to networks, and so on. This article provides for the first time an examination of the issue of reverse mentoring and of the constraints facing SMEs in this issue.
Literature is well analyzed and clearly expresses the logics of reverse mentoring and the reasons why SMEs may again confront significant difficulties in their efforts to adopt this strategy. The new model of reverse mentoring devised by the authors is original, well structured, and most importantly already tested in Brazil. Although the pilot test is based on a very small sample and can hardly provide final robust conclusions, it can however be considered as an important effort in the right direction, and a base for continuing research.
Although the article deals with SMEs, I think that the relevance of the new model to the big enterprises should at least be mentioned. The extensive literature review mostly relevant to big companies is important as a background to the subject, but at the same time demands a short reference to the new model. The adaptation of the new model to big companies naturally requires a separate study, but the authors may make a contribution by indicating a short evaluation of this option.
Reviewer 2 Report
Innovation is a vital tool for promoting the competitiveness of companies. Previous studies and policymakers alike had been looking for models to help with that.
The present study presents a relatively new model for promoting innovation, which is currently operated mainly by large companies. Although in need of assistance in promoting innovation, SMEs are not to use this model. The present study has a dual contribution.
First – it presents a comprehensive literature review of the RM model. Secondly, its main contribution is the formulation of a new RM model suitable for small businesses and offers an empirical experiment that proves its feasibility for SMEs and its effectiveness.
Minor comments and suggestions:
1) The study refers to the demand by companies for support tools to implement the new direction in the fields of managerial tools and human resources. I suggest giving more examples for both fields
2) In the section dealing with the assistance of innovation agents in promoting open innovation, I propose adding a short paragraph addressing the definition of open innovation.
3) Previous studies on SMEs show that the difficulties of SMEs in recruiting finance hinder their innovation potential. The current research shows that assistance through the proposed RM model can encourage and promote innovation among SMEs. This should be indicated in the conclusions.
4) The last paragraph that starts with; “Our recommendation is to expand the study to more cases and other countries.” should be titled recommendation for further research
5) In the recommendations for further research, I suggest exploring the effectiveness of the proposed (new) RM model for integrating university graduates into the labor market.
6) Other comments:
(1) In table 1, the left size of the parenthesis should be corrected – instead of: )Fortaleza/Periphery) it should be: (Fortaleza/Periphery). (2) In table 2, I recommend adding to the title: “and the demand for supporting tools.” Therefore, the full title should be: “Companies by size and the directions for promoting innovation and the demand for supporting tools.”
Reviewer 3 Report
Dear authors
I appreciate the work you have done. However, I am not sure whether the proposed system can work with SMEs, particularly small and micro ones.
Firstly, I am not sure if the system claimed can work with SMEs simply because SMEs have a different structure from big firms or companies. SMEs, particularly the medium ones, let alone the micro ones, are operated mainly by one or two or a few people. How is it possible to have senior people in micro or small enterprises? Primarily, these types of businesses are handled by the business owners, the decision-makers. Thus this system can not be applied in the world of SMEs.
Secondly, regarding the article's write-up and struture, I felt I am reading a story, not an essay. The writing up is inferior, particularly the introduction part. There is minimal literature in the whole article and the introduction. There is no clear research gap or problem in the study. I can not either consider it a quantitative or qualitative paper. There is no clear implication or discussion.
Additionally, the article does not seem adequate for the journal; I believe MDPI has a particular format; however, I see here that the author is following APA writing style or maybe his one style. I am wondering if the journal accepts it.
I hope my comments will not discourage you from continuing to improve the article and submitting it.
Kind regards
Reviewer 4 Report
Some common comments include:
1. Use the writing guidelines used in sustainability journals and focus on templates to be systematic according to the author's guidelines
2. There are several additional literatures that need to be completed in this manuscript
3. Clarification of Methods, Findings and Discussion
4. Improve writing Conclusion
5. Use the Reference Manager Tool
(For a more complete commentary given, look directly at the technical text)
